# Astrocytes: The Stars in Neurodegeneration?

**DOI:** 10.3390/biom14030289

**Published:** 2024-02-28

**Authors:** Katarina Stoklund Dittlau, Kristine Freude

**Affiliations:** Department of Veterinary and Animal Sciences, Faculty of Health and Medical Sciences, University of Copenhagen, 1870 Frederiksberg, Denmark; katarina.dittlau@sund.ku.dk

**Keywords:** astrocyte, neurodegeneration, reactivity, non-cell autonomous, cell autonomous, cross-disease

## Abstract

Today, neurodegenerative disorders like Alzheimer’s disease (AD), Parkinson’s disease (PD), frontotemporal dementia (FTD) and amyotrophic lateral sclerosis (ALS) affect millions of people worldwide, and as the average human lifespan increases, similarly grows the number of patients. For many decades, cognitive and motoric decline has been explained by the very apparent deterioration of neurons in various regions of the brain and spinal cord. However, more recent studies show that disease progression is greatly influenced by the vast population of glial cells. Astrocytes are traditionally considered star-shaped cells on which neurons rely heavily for their optimal homeostasis and survival. Increasing amounts of evidence depict how astrocytes lose their supportive functions while simultaneously gaining toxic properties during neurodegeneration. Many of these changes are similar across various neurodegenerative diseases, and in this review, we highlight these commonalities. We discuss how astrocyte dysfunction drives neuronal demise across a wide range of neurodegenerative diseases, but rather than categorizing based on disease, we aim to provide an overview based on currently known mechanisms. As such, this review delivers a different perspective on the disease causes of neurodegeneration in the hope to encourage further cross-disease studies into shared disease mechanisms, which might ultimately disclose potentially common therapeutic entry points across a wide panel of neurodegenerative diseases.

## 1. Introduction

Astrocytes received their name from Hungarian anatomist and histologist Michael von Lenhossék (Mihàly Lenhossék) in 1895 due to their mesmerizing star-like morphology [1]. Since then, multiple astrocytic subpopulations have been defined based on morphology, function and spatiotemporal distribution, and the portfolio is continuously growing from the emerging of single-cell and single-nuclei sequencing studies [2]. Now, it is evident that individual astrocytic subtypes might change during development and aging and in response to external stimuli in their immediate environment, which jointly changes the composition of the astrocyte population [2]. Similarly intriguing is the notion that individual astrocytes can exhibit both deleterious and protective properties in pathological scenarios and during disease progressions and that these functions might change in response to both intrinsic and external stimuli [3,4]. 

Much is already known about the astrocytes’ function during normal physiological conditions, which is an important foundation for the phenotypic understanding and extrapolation during diseases. Astrocytes are crucial for the integrity and function of the neuronal network, as they form tripartite synapses with neurons and thereby ensure activity modulation and neurotransmitter regulation through, e.g., glutamate uptake [5]. Additionally, astrocytes monitor and regulate the water flux and ion and pH homeostasis and provide structural integrity to the extracellular matrix in order to sustain an optimal environment [5]. As a key component and regulator of the selective permeable blood–brain barrier (BBB), astrocytes provide neurotrophic support, nutrients and waste removal, and together with microglia, they form the first line of defense against harmful agents [6]. In summary, astrocytes constantly cater to their surrounding neuronal population in order to maintain optimal function of the neural circuit. 

Some subgroups of cortical astrocytes are shown to be located in individual three-dimensional domains, where they are believed to adapt their gene expression profile to neurons in their residing regions [2]. This notion is based on the observation that other astrocytes from different brain or spinal cord regions are unable to compensate in case of astrocyte domain loss in rodents [7,8]. Similarly, astrocytes are shown to display region-specific transcription factors, which contributes to their heterogeneity across brain regions [9,10,11]. Different anatomical regions are affected in various neurodegenerative diseases (AD: hippocampus and entorhinal (initially) and cortex (later), PD: substantia nigra, FTD: frontotemporal lobes, ALS: motor cortex, brain stem and spinal cord, Huntington disease (HD): striatum) and might therefore correlate with regional astrocytic failure to support the neurons locally, thereby progressing the diseases [12]. Additionally, some brain regions might also be more vulnerable to intrinsic pathological changes and external toxic insults in neurodegenerative diseases or naturally occurring changes such as during aging [13]. Aging is especially a common risk factor for developing neurodegenerative diseases and has, among other functions, been shown to affect the immune response in astrocytes [14,15,16]. Nonetheless, further research is required to clarify if astrocyte region-specific heterogeneity and dysregulation is a driving mechanism in neurodegeneration.

The functional complexity is mirrored in pathology. As a natural response to injury, infection or disease, astrocytes become reactive through morphological, transcriptional and functional transformations [17]. From primarily being tissue-embedded and non-motile, reactive astrocytes abandon certain neuronal supportive obligations in favor of an inflammatory activation through BBB remodeling, cytokine secretion and border formations [18,19]. These reactive astrocytes are believed to adapt based on the pathologic condition and can thus take on both neurosupportive and neurotoxic roles [3]. Such adaptations are highly complex and favor the concept of astrocytic “sub-states”, where specific pathologic conditions result in specific reactive astrocytes [20,21,22]. These astrocytic sub-states are likely modulated by the homeostatic need of their residing region and the nature of the pathological condition and might be dynamic during different disease states, thereby moving across a “reactive astrocyte spectrum” [21,22]. Importantly, astrocyte activation as an adaptive physiological response to insults to reestablish homeostasis and protect neuronal function should not be confused with disease-induced abnormal or chronic astrocyte reactivity observed in neurodegenerative diseases [20]. In the latter, the response often starts as being neuroprotective but eventually develops into toxicity, further contributing to cellular stress and neurodegeneration. Here, the underlying mechanisms are not well understood, and the widespread astrocyte heterogeneity only complicates the investigation [22,23]. However, despite the potential myriad of astrocyte subpopulations and sub-states, it is evident that multiple pathways are in fact common across a wider panel of neurodegeneration [24], which argues for a cross-disease investigative approach. 

## 2. Neuroprotection versus Neurotoxicity 

Neurodegenerative diseases affect more than 60 million people worldwide, and currently, no cure exists for any of the major groups such as AD, PD, FTD, ALS and HD. Despite being vastly different in genetic background, age of disease onset, clinical representation and phenotypic involvement, these neurodegenerative diseases have several hallmarks in common (Figure 1). The first and major one is their definition as proteinopathies. In each of these diseases, mutant proteins undergo pathological conformational changes, which results in accumulation and the formation of intracellular and/or extracellular aggregates (Figure 1) [12]. For AD, mutant amyloid-β (Aβ) aggregates into extracellular amyloid plaques and phosphorylated tau causes intracellular neurofibrillary tangles. For HD, mutated huntingtin forms intracellular aggregates, while α-synuclein accumulates in Lewi bodies in PD. In ALS, mutant TDP-43, FUS or SOD1 protein are shown to accumulate and aggregate, as for TDP-43 and/or tau in FTD, resulting in aggregation and neurofibrillary tangles, respectively. Many of these pathological intracellular protein buildups are found in the glia population as well [25,26,27,28,29,30,31]. This is likely an important contributing factor to the second common hallmark, which is neuroinflammation (Figure 1) [2,32].

Neuroinflammation can manifest in several ways. As mentioned previously, reactive astrocytes are shown to take on both neuroprotective and neurotoxic roles during the disease progression of neurodegenerative diseases [23,33]. Many aspects are still not well understood, but the general consensus is that the initial reactive response is considered neuroprotective but eventually and with continuous reactivity develops into a chronic and toxic inflammation, leading to degeneration of the neurons in the area. This development has detrimental consequences to the environment, including the neuronal network, which leads to the final common hallmark of neurodegenerative disease: neuronal toxicity and cell death (Figure 1). Often times, the astrocytic shift from neuroprotection to neurotoxicity happens gradually as the noxious stimuli accumulate, which ultimately escalates the disease progression [34]. As such, both predominantly neuroprotective and predominantly neurotoxic reactive astrocytes might reside in various regions of the brain at similar time points during the disease progression [13,23,33,35]. 

Examples in favor of a reactive astrocytic phenotype with neuroprotective characteristics can be observed in multiple neurodegenerative diseases. In AD, amyloid plaque deposition increased in an *APPswe/PS1ΔE9* transgenic mouse model when astrocyte reactivity was reduced through glial fibrillary acid protein (GFAP)/Vimentin double knockout [36]. A similar acceleration of the disease progression was seen upon reactive astrocyte abolishment in prion disease [37]. In contrary, JAK2-STAT3 activation in reactive astrocytes in the *Hdh140* HD transgenic mouse model was shown to induce proteostasis, thereby reducing the mutant huntingtin aggregation burden [38]. Additionally, increased clusterin (apolipoprotein J) secretion from astrocytes was able to rescue synaptic deficits and ameliorate Aβ neuropathology in the 5xFAD transgenic mouse model of AD [39]. Reactive astrocytes accumulate around amyloid plaques in human post-mortem tissue [13,40,41,42]; however, it is unknown if this is due to a reactive astrocytic attempt to constrain the toxicity or if amyloid plaques are driving reactivity in adjoining astrocytes [12]. In favor of neurotoxic reactivity, Liddelow and colleagues showed how some astrocytes encompassed deleterious properties, resulting in neurotoxicity after acute CNS injury in rodents [3]. This “A1” neurotoxic astrocyte sub-type was induced by microglial cytokine secretion and showed a specific transcriptional pattern while being unable to support synapse integrity. Interestingly, markers of A1 reactivity were also found in astrocytes in post-mortem samples from AD, PD, HD, ALS and multiple sclerosis (MS) patients, suggesting that neurotoxic astrocytes are present across a wide panel of neurodegenerative diseases [3,4].

Despite these cases of either predominant neuroprotection or predominant neurotoxicity, they very likely do not describe the full picture. For example, NFΚΒ pathway activation in astrocytes has been shown to mediate an initial neuroprotective astrocytic function with microglial anti-inflammatory activation, leucocyte infiltration and prolonging of disease onset during the pre-symptomatic phase of ALS [43]. However, the same continued astrocytic NFΚΒ activation later develops into a pro-inflammatory response, which accelerates the disease progression [43]. This argues for a dynamic transition between neuroprotection and neurotoxicity and, as a result, we must look at astrocytic contributions to neurodegenerative diseases with caution and avoid the temptation for simplistic categorization. 

Many other cell types in the brain and spinal cord (neurons, microglia, endothelial cells, etc.) contribute to neuroinflammation and have shown to precede, initiate or drive astrocyte reactivity [3,44,45,46]. Similarly, astrocytes have been shown to govern microglial activation, and a large array of studies document intrinsic activation of monocultured astrocytes independent of their environment [43,47,48]. In the next part of this review, examples of shared astrocytic non-cell autonomous and cell autonomous mechanisms currently described across multiple neurodegenerative diseases will be provided.

## 3. Non-Cell Autonomous Mechanisms 

Astrocytes are able to modulate their surroundings through secretion of many types of proteins. Through transsynaptic adhesion proteins or receptors modulations, astrocytic released molecules such as hevin (SPARCL1), thrombospondins (THBS1/2), glypicans (GPC4/6), transforming growth factor β1 (TGF-β1) and brain-derived neurotrophic factor (BDNF) are able to promote the formation and maturation of excitatory synapses [18,49]. In addition, astrocyte-secreted tumor necrosis factor-α (TNF-α) aids in increasing neuronal activity, as it promotes the presence of α-Amino-3-hydroxy-5-methyl-4-isoxazolepropionic acid (AMPA) receptors on excitatory synapses and diminishes gamma-aminobutyric acid (GABA) receptors on inhibitory synapses [50]. Some secreted proteins like TGF-β1 have multiple functions, as it likewise promotes the formation of inhibitory synapses in the central nervous system [49]. 

One of the first known mechanisms of astrocyte non-cell autonomous effects in neurodegeneration is the apparent change in their secretome. Astrocytic releases of cytokines, chemokines and interleukins jointly increase the inflammatory response through astrocytic self-activation, microglial stimulation as well as periphery leukocyte involvement [6]. Simultaneously, chronic exposure to this secretory stimulation is causing several downstream pathologies, ultimately resulting in neurotoxicity and cell death. 

In MS, astrocytes secrete chemokines such as CXCL10, CXCL12 and CCL20, which promote periphery immune cell infiltration [51,52,53], while cytokine secretion involving IL-6, TNF-α and IFN-γ is prominent in PD [29,54]. In AD, multiple complement effector proteins such as complement factor 3 (C3) are secreted directly or in exosomes in addition to IL-6, TNF-α and IL-1β, thereby driving a disease-stage-dependent inflammatory response involving microglial Aβ phagocytosis [55,56]. More specifically, Aβ is shown to drive an NFΚΒ-dependent and chronic astrocytic release of C3, which subsequently hampers microglia’s ability to phagocytose Aβ aggregates [57]. Additionally, astrocytes might also contribute to the amyloid burden in AD by increased secretion of Aβ [58], which could trigger additional astrocytic reactivity and self-activation [59]. Many of these inflammatory molecules are likely secreted by astrocytes in an attempt to evoke distinct downstream pathways thought to alleviate the pathology; however, chronic exposure evidently causes considerable dysfunction. As an example, we previously showed how FTD astrocyte-conditioned medium containing IL-6, IL-8 and IL-13 could inhibit axonal outgrowth of control neurons [60]. In ALS, large efforts have produced profound insight into the dysregulated astrocyte secretome. In both familial and sporadic ALS, multiple studies document a dysregulated astrocytic release of cytokines like TGF-β, TNF-α, IL-6 and IL-1β in addition to prostaglandins, inorganic polyphosphate, nitric oxide (NO), reactive oxygen species (ROS) and miRNA-containing extracellular vesicles, which cause abnormal microglial function and motor neuron toxicity [61,62,63,64,65,66,67,68,69,70,71,72,73,74,75]. This astrocyte secretome dysfunction in ALS has been shown to impair many key motor neuron functions involving autophagy [69,76,77], neurite outgrowth and length [62,69,73] and neuromuscular junction innervation [73,78,79], ultimately resulting in accelerated protein aggregation, excitotoxicity, cellular stress and motoric/respiratory degeneration and failure [48,79,80,81,82,83,84,85]. 

## 4. Cell Autonomous Mechanisms

A constant stream of evidence has firmly documented that astrocytes not only contribute to neurodegenerative disease pathology and progression through non-cell autonomous mechanisms but in fact also display multiple intrinsic pathological phenotypes. In AD, PD, ALS, FTD, MS as well as prion disease, astrocytes harbor increased expression of GFAP and C3, which are thought to be important (albeit not exclusive) markers of astrocyte reactivity encompassing a neurotoxic phenotype [3,37,60,78,86]. These protein expressions are part of a general observation of altered transcriptomic and proteomic levels, which largely result in a downregulation of astrocytic support functions such as ion and cholesterol homeostasis, glutamate uptake and synapse integrity and upregulation of pro-inflammatory pathways such as JAK-STAT3, NFAT and NFΚΒ with a resultant astrocytic toxic gain of functions [38,40,67,69,70,73,87,88,89,90,91,92,93,94]. In the following section, the cell autonomous mechanisms in astrocytes, which are common across multiple neurodegenerative diseases, are highlighted (see also Figure 2).

### 4.1. Communication

Unlike neurons, astrocytes are non-excitable and therefore have developed other means of communication. Besides cytokine, chemokine and interleukin releases mentioned above, astrocytes use calcium waves, transmitter release and gap junction couplings as a means of communication to maintain network homeostasis [49].

#### 4.1.1. Calcium

Astrocytic calcium transients are highly dynamic. The “waves” are projected intra- and intercellularly through calcium-permeable ion channels and receptors, including various neurotransmitter receptors [5]. Upon neurotransmitter binding, a receptor-specific intracellular calcium signal is evoked, which allows astrocytes to distinguish between, e.g., glutamatergic and cholinergic synaptic activity [95,96]. Individual astrocytic subpopulations display unique calcium waves, which further contributes to their heterogeneity and complexity [97]. 

In multiple neurodegenerative diseases including AD [42,98,99,100,101,102], ALS [103,104], HD [105,106,107] and PD [29,108], calcium dyshomeostasis is observed. The dysregulated calcium response is often detected early in disease progression and might therefore contribute to the pathologically altered neuronal synapse activity [102,105,107,109,110,111]. For example, oligomeric forms of Aβ peptide were shown to drive astrocytic calcium hyperactivity early in AD disease progression, which consequently triggered glutamatergic hyperactivity in adjacent neurons [109,110]. Calcium is primarily stored in the endoplasmic reticulum (ER) within astrocytes but can also be imported into the cells through AMPA and N-methyl-D-aspartate (NMDA) receptors or through voltage-gated channels [49]. In ALS and PD, abnormal intracellular calcium dynamics with excess ER accumulation and storage release is observed, ultimately contributing to ER stress [29,103,104]. Both genetic PD mutations and drug-induced depletion of dopamine transmission mimicking PD have shown to increase astrocytic calcium excitability [29,108]. In the R6/2 transgenic mouse model of HD, spontaneous calcium signals and storage capacity are reduced, while evoked calcium responses are increased [105].

#### 4.1.2. Intercellular Communication

In addition to receiving neurotransmitter information, calcium transients can release transmitters such as glutamate, GABA, D-serine and ATP from astrocytic processes, which contribute to the regulation of neuronal synapse excitability and plasticity [49,95]. In pathological conditions, the release of these transmitters from both neurons and astrocytes is dysregulated, and studies show that excess release causes a continuous self-activation, leading to cytotoxicity [42,99,112]. This phenomenon is observed in several models of PD and AD and is triggered by excessive astrocytic calcium-dependent release of glutamate and D-serine [110,111,113]. Similarly, astrocytic release of ATP can bind purinergic P2Y receptors on microglia, thereby modulating their phagocytic functions and release of cytokines further driving inflammation [114,115]. Astrocytic self-activation is also enhanced by the release of astrocytic TNF-α and prostaglandin, which, through calcium signaling, can promote additional “gliotransmitter” release [112]. 

The astrocyte–neuron communication can further be impaired through other means such as extracellular vesicle (EV) production. In *LRKK2*-mutant astrocytes, the EV biogenesis is altered, causing an accumulation of PD-related proteins within multivesicular bodies [116]. These astrocyte-secreted EVs are internalized by dopaminergic neurons and are linked to an astrocytic failure of providing neurotrophic support [116]. Furthermore, the previously mentioned C3-containing exosomes are released by AD astrocytes, consequently driving an inflammatory response and being linked to reduced neurite outgrowth [55,117]. 

#### 4.1.3. Gap Junctions

Astrocytes not only communicate via neurotransmitters but also through gap junction couplings. Connexin-43 (Cx43) is the predominant connexin protein, which constitutes hemichannels and gap junctions in astrocytes [118,119,120]. Through these, diffusion of ions, metabolites, miRNAs and second messengers is facilitated, in addition to the important mitigation of calcium waves [121]. Therefore, gap junctions are key components in astrocyte networks, contributing to synapse activity modulation and homeostatic buffering. In ALS [69,122,123], AD [42,124] and PD [108], Cx43 is abnormally elevated, which causes increased gap junction coupling and hemichannel activity, resulting in calcium hyperactivity, neuronal excitability and cell death. Astrocytes also form collaborative glial networks with oligodendrocytes through their Cx43/Cx47 gap junction connections, which are important for their coordinated cross-talk [125]. In MS, oligodendrocytes and astrocytes lose their communication through decreased Cx43/Cx47 gap junction expression, consequently promoting demyelination and inflammation [126,127,128]. Lack of Cx43/Cx47 gap junction couplings is likewise found in AD [129].

### 4.2. Tripartite Synapse: Neurotransmitter Regulation and Synapse Function

Astrocytes are highly involved in neuronal synapse plasticity and function through their perisynaptic process ensheathment of neuronal synapses [130]. This tripartite synapse collaboration between pre- and postsynaptic neurons and astrocytes ensures optimal neuronal firing by continued astrocytic removal and recycle of excess neurotransmitters from the intersynaptic space [131,132]. In neurodegeneration, this function is impaired. Due to the downregulation of key astrocyte receptors, excitatory amino acid transporters 1 and 2 (EAAT1 and EAAT2), astrocytes fail to properly manage the uptake of the neurotransmitter glutamate. Glutamate is the main excitatory neurotransmitter in the brain, and a large part of its uptake from the synaptic cleft appears through EAAT1/2 receptors on astrocytes [133,134,135]. Persistent glutamate exposure is believed to cause excessive neuronal firing and abnormal neuronal calcium influx, which ultimately results in severe neuronal excitotoxicity [131,134,136,137]. Lack of EAAT1/2 receptors and consequent glutamate excitotoxicity is a common phenomenon in AD [138,139,140,141,142], PD [143,144], ALS [48,83,145,146] and HD [27,28,105]. Additionally, downregulation of astrocytic glutamate receptors drives abnormal microglial pruning and phagocytosis of hippocampal glutamatergic synapses in AD [147]. In HD, structural pathology, mediated by lack of astrocytic engagement of neuronal synapses, enables the hyper-excitability [148]. AD astrocytes likewise have perturbed glutamine metabolism and supply, which affects the GABA synthesis [149,150,151,152]. GABA is the main inhibitory neurotransmitter in the central nervous system, and lack thereof contributes to the excitatory imbalance [5]. In addition to neurotransmitter uptake, astrocytic release of various transmitters mentioned above is additionally important for synapse activity regulation. Abnormal gliotransmission has thus been shown to affect the synaptic transmission in PD and AD, consequently contributing to synapse loss and excitotoxicity [110,113,137,153,154,155]. Similarly, astrocytic release of C3 can bind to neuronal C3 receptors and hamper their synaptic density and dendritic morphology [156]. 

Astrocytes control the ion homeostasis through multiple ion channels, which is crucial for maintaining synapse functionality [157]. Potassium (K^+^) buffering is a key mechanism affected in neurodegeneration. Under physiological conditions, astrocytes regulate K^+^ levels in the extracellular space through clearance via K^+^ channels such as the main astrocytic Kir4.1 sub-type [49]. Through these mechanisms, astrocytes can modulate neuronal depolarization and thereby excitability [158]. In HD [26] and ALS [159], astrocytic Kir channels are downregulated, which hampers the K^+^ buffering and clearing, consequently causing increased extracellular K^+^ levels, overall leading to neuronal excitotoxicity. Besides through Kir.4.1 channels, K^+^ is released through EAAT2 and as mentioned previously, the astrocytic release and uptake of transmitters is modulated by calcium transients. This interconnected relationship between ion homeostasis, calcium dynamics and gliotransmission is therefore crucial for optimal neuronal function, and any dysregulation of one mechanism might consequently disrupt the others. 

### 4.3. Mitochondrial Function

Mitochondrial dysfunction, ROS secretion and oxidative stress are common findings in neurodegeneration. In AD, mitochondrial dysfunctions are predicted as an early astrocytic phenotype consequently driving astrocyte reactivity [23], and in PD, maintenance of mitochondrial DNA is correlated with reduced mitochondrial respiration [29]. Astrocytic accumulations of α-synuclein in PD have been linked to damage within the mitochondrial structure, consequently lowering the total ATP level as well as causing disruption of the fission and fusion dynamics [160,161]. In ALS, mitochondria display decreased membrane potential and a compromised oxygen consumption, possibly correlated with a lower secretion of antioxidants [85,162]. Astrocytes express and release antioxidants as a part of their function in regulating the redox balance through removal of ROS in order to prevent oxidative damage of neurons [163]. Astrocytic nuclear factor erythroid 2-related factor 2 (Nrf2) transcription factor is a key regulator of antioxidant, detoxification and proteostasis pathways and might therefore be an important mediator in neurodegeneration [163]. In amyloid and tau pathology models of AD, pre-emptive activation of astrocytic Nrf2 was shown to be neuroprotective by attenuating the aggregation burden and slowing disease progression [33]. Similar observations were seen by overexpressing Nrf2 in astrocytes in *SOD1*-ALS mice [164] and in α-synuclein-mutant mice in PD [165], which advocates for a common therapeutic mechanism in neurodegeneration. Interestingly, *Nrf2* target gene levels are in fact increased in AD but might appear too limited or too late in the disease progression to make a considerable neuroprotective difference [33,166]. 

### 4.4. Energy Metabolism

Neurons rely on astrocytes for nutritional support to meet their optimal energy consumption. Through key mechanisms such as the lactate shuttle, astrocytes convert glucose or glycogen to lactate via aerobic glycolysis and provide the lactate to adjacent neurons via monocarboxylate transporters [167]. In the neurons, the lactate is incorporated into the oxidative cycle for ATP production [167]. Lactate is also shuttled directly to neurons from the bloodstream via the astrocytes’ perisynaptic process engagement with the vasculature [168]. In ALS, astrocytic intra- and extracellular levels of lactate are decreased, possibly due to diminished lactate production and/or transport [169,170]. Similarly, AD astrocytes have decreased glycolysis and secretion of lactate to the environment [101]. Lack of lactate production causes metabolic dysfunction and failure of adequate neuronal energy supply. Additionally, lactate production is coupled to glutamate transporter activity, and their shared impairment might therefore further hamper the energy metabolism in neurodegenerative disorders [12]. 

A widespread dysregulation of metabolites is generally observed in neurodegenerative diseases. Astrocyte metabolic proteomics is increasingly enriched during the commencement and disease progression of AD, with markers expressing both neuroprotective and neurotoxic phenotypes [24]. More specifically, AD astrocytes display augmented glycolytic flux and reduced glycogen storage [142]. In PD, astrocytes have altered polyamine and phospholipid levels [29], and in ALS, astrocytic metabolic dysfunction with compromised adenosine, fructose and glycogen metabolism is observed [171,172]. 

Finally, astrocytes are the main synthesizers and suppliers of apolipoprotein E (ApoE), which ensures sufficient transport of cholesterol to neurons [49]. Cholesterol is an essential lipid in cell membranes and presynaptic vesicle formation and is therefore crucial for the integrity and function of synapses. One of the main risk factors of late-onset AD is the human isoform *APOEε4* variant [173,174,175,176]. Carriers are shown to have pathologically altered glucose metabolism, lactate production, cholesterol homeostasis and calcium dynamics, resulting in AD [98,177,178,179,180]. Dysfunctional cholesterol and lipid metabolism are also found in HD [35,181,182] and PD [183,184] and appear to be a general mechanism of reactive astrocytes, potentially influencing many neurodegenerative diseases [185].

### 4.5. Clearance of Protein Aggregates

Astrocytes have been shown to contribute to the insufficient clearance of protein aggregates in neurodegeneration due to failure of the glymphatic system. Under physiological conditions, astrocytes regulate water flux as well as removal of metabolic waste products from the brain interstitium to and from the perivascular space through their Aquaporin-4 (AQP4) water channels [5]. Additionally, through the synergistic collaboration between AQP4 and Kir4.1 channels, osmohomeostasis is maintained in the brain [186,187,188]. During disease progressions, astrocytes remove Aβ peptides and mutant huntingtin through the glymphatic system [189,190]. However, due to the downregulations of Kir4.1 and AQP4, this clearing mechanism is impaired, consequently accelerating the aggregation burden [26,191,192]. Additionally, astrocytic ApoE is normally involved in Aβ clearance, but in *APOEε4* carriers, this function appears to be compromised, which likewise contributes to the enhanced Aβ aggregation [179,193]. 

Secondly, astrocytes contribute to clearance of proteins through their intracellular lysosomal pathways. This mechanism is affected in PD, where both α-synuclein oligomers as well as mutations in the *LRRK2* gene interfere with the astrocytic clearance of the aggregation burden [160,194]. Neuronal α-synuclein is taken up by astrocytes from the extracellular space via endocytosis as well as transmitted directly from neurons [93,195]. Similarly, aggregates are transferred between astrocytes via nanotubules [161]. This increased inclusion burden aided by overload/stress-induced insufficient lysosomal degradation further triggers cellular toxicity and reactivity [93,161]. More specifically, α-synuclein is shown to alter the lysosomal morphology, distribution and function by alkalinization and decrease activity of lysosomal proteases in neuronal cells and idiopathic PD brains [196,197], which correlates with similar observations of disrupted lysosomal proteolysis recently observed in astrocytes in an early-onset PD model [198]. Astrocytic engulfment of monomeric tau from the extracellular environment has been observed in tauopathies [199], and the endo-lysosomal dysfunction in astrocytes is likewise predicted as an early disease phenotype in AD [23,200]. 

Finally, microglial and astrocytic cross-communication is important for the optimal clearance of Aβ and α-synuclein aggregates [201]. Intercellular miscommunication or toxic reactivity might therefore hamper this essential function, thereby accelerating disease progressions. 

### 4.6. Autophagy

Autophagy is a vital cellular process, which facilitates the degradation and recycling of intracellular components such as damaged organelles and protein aggregates [202]. Several conditions like starvation, stress and pharmacological treatment can modulate autophagy to facilitate a faster release of important nutrients in the reestablishment of cellular homeostasis [203]. Autophagy is a multi-step process, where formation of large double-membrane vesicles termed autophagosomes bind and enclose cargo destined for degradation through ubiquitin-P62-LC3 protein interactions [202]. The autophagosomes fuse with late endosomes and finally with lysosomes, and the lysosomal enzymes degrade the vesicular content for reuse [204]. 

Mutations in *LRRK2* are a common cause of genetic PD [205]. *LRRK2* contributes, among many functions, to the phosphorylation of various proteins from the Rab family as well as P62, thereby filling a prominent role in autophagy initiation and vesicle transport [206]. Consequently, *LRRK2*-mutant astrocytes display impaired autophagy, possibly linked to progressive accumulation of α-synuclein as mentioned above [207]. Additionally, α-synuclein oligomers have been shown to interfere with the autophagosome–lysosome fusion, consequently halting the autophagic flux in astrocytes [161]. This degradation impairment results in insufficient turnover of damaged mitochondria [161]. Moreover, in ALS/FTD-relevant *C9ORF72*-mutant models, accumulation of P62 is present in astrocytes [208], and we previously showed how human induced pluripotent stem cell-derived FTD astrocytes displayed insufficient autophagy, consequently perturbing the mitochondrial turnover [60]. This lack of mitophagy resulted in augmented mitochondrial fusion, impaired mitochondrial respiration and glycolysis, increased ROS and stress granules formation, which overall triggered astrocyte reactivity and cytokine secretion [60].

### 4.7. Blood–Brain Barrier 

Through close interaction with capillary endothelial cells and pericytes, astrocytes function as efficient gatekeepers of the central nervous system through regulation of the integrity and permeability of the BBB [6]. Additionally, neurons rely on astrocytes for shuttling of nutrients and metabolic substrates from the blood stream through the BBB [6], and with calcium-dependent release of vasodilators or vasoconstrictors, astrocytes control the blood flow depending on the neuronal energy demand [209]. As mentioned previously, AQP4 levels are changed in various neurodegenerative diseases, which also contribute to the compromised BBB [13,210]. Therefore, cerebrovascular deterioration in AD could be correlated with astrocytic calcium hyperactivity [42].

In ALS, astrocytes contribute to BBB disruption and pericyte loss, which allow periphery leukocyte infiltration and a disturbance of the homeostatic environment [43,211,212,213,214]. In AD, *APOEε4* carriers display pericyte degeneration due to insufficient astrocyte suppression of proinflammatory pathways, consequently damaging the BBB [215,216]. Additionally, the barrier function of BBB tight junctions is impaired when astrocytes carry the *APOEε4* isoform [217], and astrocytes modulate leukocyte infiltration due to Aβ exposure [218]. In HD, astrocytes trigger endothelial cell proliferation and pericyte damage, which disrupts the vascular function [219]. Finally, in PD, reactive astrocytes fail to support vessel formation and barrier integrity [220]. 

### 4.8. Glial Border Formation

Astrocytes are crucial for the integrity of the stroma in the central nervous system and provide important structural support [5]. Through their secretion of molecules such as proteoglycans, astrocytes contribute to the extracellular matrix (ECM) structure surrounding synapses and within the synaptic cleft. This is important for capturing nutrients and growth factors as well as acting as a diffusion barrier for neurotransmitter concentration buffering [49]. During neurodegeneration, subpopulations of astrocytes initiate a border formation, which primarily functions to contain the injury. However, emerging evidence point towards a rising role of abnormal scarring in these disorders [221]. Here, ECM molecules promote a pro-inflammatory signal mediated by microglia and infiltrating macrophages, which further drives the astrocytic reactivity and matrix release [221]. Conversely, glial border formation is shown to be beneficial in its formation of “glial bridges”, along which axonal regrowth is permitted if appropriately guided by growth factors [222]. Lack of astrocytic neuro-supportive functions likely influences this mechanism in neurodegenerative diseases [34].

Chondroitin sulfate proteoglycans (CSPGs) are central components of glial border formation and mediated by reactive astrocytes in MS lesions [223]. Although this physical barrier aids in the containment of the lesion, it also prevents axonal outgrowth and remyelination [223]. As a consequence, excessive glial border formation by astrocytes might distort the tissue architecture, thereby prolonging the disease course of MS. Similarly, in ALS, CSPG accumulation is present at the site of motor neuron degeneration, and excessive and abnormal CSPG receptors are also found on the surface of reactive astrocytes, further contributing to their self-activation and the overall impairment of the homeostatic environment [224,225]. Additionally, chronic release of TGF-β promotes excessive fibrosis in ALS [226] and acts as an upstream regulator of CSPG secretion [227,228]. In AD, astrocytic proteoglycans interact with Aβ plaques, consequently promoting aggregation and inhibiting clearance [229].

Finally, the STAT3 pathway in astrocytes is an important modulator of glial border formation after spinal cord injury [230]. Abnormal expression of the STAT3 pathway in multiple neurodegenerative disease could likely contribute to border formation imbalances. Similarly, the damage to the BBB generally observed in neurodegeneration permits the infiltration of mesenchymal cells and blood proteins, which mediates excessive scarring [221]. 

## 5. Conclusions

Astrocyte physiology and pathology are complex. It is fascinating to speculate how many sub-types and sub-states exist and how astrocytic adaptability might drive neurodegenerative disease progressions. Despite their profound heterogeneity, astrocytes as a cell group share common pathological characteristics across a broad spectrum of neurodegeneration. Here, we have highlighted the disease commonalities in some of the most common mechanisms, but many more are likely present, and further research is undoubtedly warranted. What we learn from one disease could potentially be extrapolated to others, ultimately benefitting more patients. Current treatment regimens largely follow the dominant field of neurocentric studies, but this neuronal favoring has unfortunately resulted in many unsuccessful clinical trials. Given the paramount role of astrocytes in ensuring optimal neuronal function and survival, these glial cells constitute important therapeutic targets in drug development. With this review, we encourage future studies in astrocyte mechanisms with a cross-disease and cross-model investigative approach in an effort to lower the variability and potentially disclose new and shared therapeutic targets for multiple neurodegenerative diseases. 

## Figures and Tables

**Figure 1 biomolecules-14-00289-f001:**
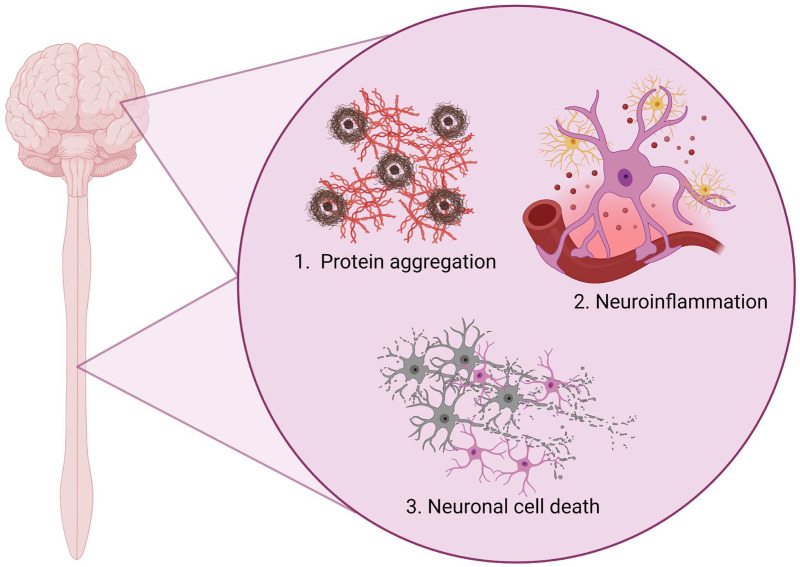
Common disease hallmarks in neurodegeneration. Figure is created with biorender.com (accessed on 28 January 2024).

**Figure 2 biomolecules-14-00289-f002:**
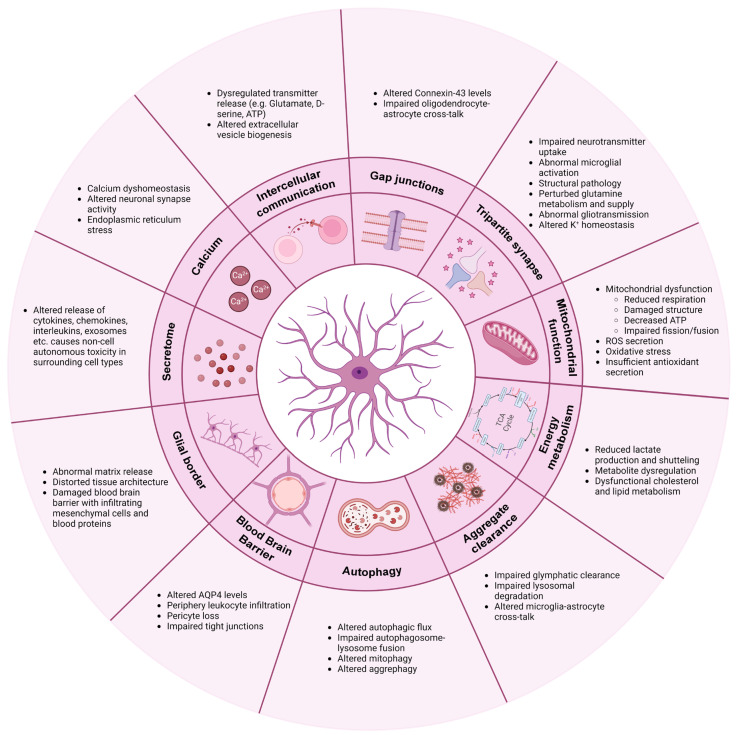
Common mechanisms involved in astrocytic pathology in neurodegenerative diseases. Figure is created with biorender.com (accessed on 28 Janurary 2024).

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
