# Peer review of "Astrocytes: The Stars in Neurodegeneration?"

_biomolecules, 2024, doi:10.3390/biom14030289_

Round 1

Reviewer 1 Report

Comments and Suggestions for Authors

Summary: The present manuscript systematically details the cell autonomous and non-cell autonomous mechanisms by which astrocytes contribute to several neurodegenerative diseases ranging from AD to HD. After a brief overview characterizing astrocyte function and heterogeneity in the CNS, the authors discuss the dual neuroprotective/neurotoxic roles observed in astrocytes in various human and transgenic murine models, with an emphasis on the temporal dynamism of astrocyte states as a function of disease progression. The authors then aggregate evidence for the dysregulation of multiple astrocyte mechanisms that to varying extents are shared between seemingly disparate neurodegenerative disorders.

Overall assessment: The authors effectively identify the major known mechanisms by which astrocytes may contribute to various neurodegenerative conditions and provide sufficient evidence underscoring previously understudied glial mechanisms of disease pathogenesis and progression. Given the inherently broad scope of this work, the reviewer encourages the authors to carefully examine the organization of the presented data to ensure that readers do not interpret them as fragmented findings from merely tangentially related studies. Moreover, certain segments of the work appear to preferentially feature AD/FTD, perhaps due to data availability; the reviewer nonetheless requests that the authors consider expanding their discussion on the other neurological disorders identified, as this will likely aid in the segmentation/organization of the manuscript.

Lines 51-61: The manuscript intriguingly discusses astrocyte heterogeneity and their role in neurodegeneration but misaligns evidence on cortical vs. spinal cord astrocytes to claim neocortical differences. The cited study, focusing on radial glia response to spinal insult, inadequately supports neocortical astrocyte heterogeneity. A more precise citation addressing cortical astrocyte differences is essential. Furthermore, asserting region-specific astrocyte dysregulation in neurodegenerative diseases without direct evidence may be premature. The manuscript would benefit from emphasizing the need for further research to robustly establish these connections and clarify the nuanced roles of astrocytes in neurodegeneration.

Lines 115-116:  This statement requires multiple citations, or a single reference to a relevant review paper.

Lines 177-200: While this passage succeeds in providing aggregated evidence of non-cell autonomous astrocytic mechanisms involved in various neurodegenerative diseases, heavy emphasis is placed on AD and FTD relative to the other conditions mentioned (e.g., PD, HD, MS, etc.). To address this, the authors should aim to provide a more balanced overview by dedicating similar lengths of discussion to each disease. This could involve expanding sections on MS, PD, FTD, and ALS to include more detailed descriptions of astrocyte contributions, like the depth provided for AD.

Moreover, the presentation of evidence across different conditions seems fragmented, which might hinder the reader’s ability to follow the complex interplay of astrocyte contributions to these diseases. To improve the organization and readability of the passage, the authors could consider restructuring the information based on thematic or mechanistic categories rather than by disease. For example, sections could be organized around specific astrocyte functions or secretions (e.g., cytokine secretion, chemokine promotion of immune cell infiltration, contributions to protein aggregation, and effects on neuronal and synaptic functions) and then discuss how these mechanisms play out across the different diseases. Alternatively, the authors can simply split the provided evidence into separate paragraphs based on disease, implementing subsection headings if necessary.

Lines 251-252: Care should be taken here to avoid implying that neurodegenerative diseases are overwhelmingly or even more commonly characterized by the excess release of the neurotransmitters listed. Both excess release and inhibited release of neurotransmitters have been implicated in the pathology of neurodegenerative diseases, with the specific patterns varying by diseases and within different stages or regions of the brain in the same disease. Please amend the phrasing of this sentence accordingly.

Lines 336-345: While the passage correctly identifies Nrf2 as a critical regulator and its potential therapeutic implications, expanding on how Nrf2 activation in astrocytes specifically influences neuronal survival and disease progression could offer deeper insights – for example, discussing the downstream effects of Nrf2 activation, such as upregulation of antioxidant response element (ARE)-driven genes, would provide a more comprehensive view.

Lines 397-398: This paragraph would benefit from additional detail regarding the mechanism(s) of lysosomal insufficiency. Is the inhibition of lysosomal degradation referenced here related to inadequate lysosomal acidification?

Line 416: Additional information about the function of LRRK2, either here, or in the previous section, would clarify the significance of its mutation and its relation to autophagy dysregulation and impaired astrocytic protein clearance.

Figure 2: The authors carefully distinguish between non cell autonomous and cell autonomous mechanisms of astrocytic contributions to various neurodegenerative diseases in the body text – this distinction/categorization should be reflected in this figure.

A few grammatical mistakes have been noted throughout the manuscript and the reviewer simply recommends the authors to do a thorough proof read.

Author Response

We would like to thank the reviewers for their thorough evaluation of our manuscript and for their constructive comments. Their input has without a doubt improved our manuscript. Please find the answers to the individual comments below. In addition to the changes outlined below, we have changed the phrasing “glial scar formation” to glial border formation” to accommodate the updated terminology in the field. All changes have been marked with track-changes in the word-document.

Reviewer 1:

Summary: The present manuscript systematically details the cell autonomous and non-cell autonomous mechanisms by which astrocytes contribute to several neurodegenerative diseases ranging from AD to HD. After a brief overview characterizing astrocyte function and heterogeneity in the CNS, the authors discuss the dual neuroprotective/neurotoxic roles observed in astrocytes in various human and transgenic murine models, with an emphasis on the temporal dynamism of astrocyte states as a function of disease progression. The authors then aggregate evidence for the dysregulation of multiple astrocyte mechanisms that to varying extents are shared between seemingly disparate neurodegenerative disorders.

Overall assessment: The authors effectively identify the major known mechanisms by which astrocytes may contribute to various neurodegenerative conditions and provide sufficient evidence underscoring previously understudied glial mechanisms of disease pathogenesis and progression. Given the inherently broad scope of this work, the reviewer encourages the authors to carefully examine the organization of the presented data to ensure that readers do not interpret them as fragmented findings from merely tangentially related studies. Moreover, certain segments of the work appear to preferentially feature AD/FTD, perhaps due to data availability; the reviewer nonetheless requests that the authors consider expanding their discussion on the other neurological disorders identified, as this will likely aid in the segmentation/organization of the manuscript.

We would like to thank the reviewer for this positive evaluation. As the reviewer rightfully states, larger emphasis has been placed on studies describing astrocytic pathologies in AD. This is, as the reviewer points out, due to the larger data availability in these more heavily studies fields. The aim of this review is not to include every study in every field, as we believe it would exhaust the reader, but merely give an overview of examples from different diseases, which jointly argues for common pathological mechanisms. With this aim in mind, we believe the organization favours the apparent balance in data availability while still justifying mechanistic overlap across various neurodegenerative diseases. Additionally, the changes to the manuscript outlined below have also favoured additional information on e.g. PD.

Lines 51-61: The manuscript intriguingly discusses astrocyte heterogeneity and their role in neurodegeneration but misaligns evidence on cortical vs. spinal cord astrocytes to claim neocortical differences. The cited study, focusing on radial glia response to spinal insult, inadequately supports neocortical astrocyte heterogeneity. A more precise citation addressing cortical astrocyte differences is essential. Furthermore, asserting region-specific astrocyte dysregulation in neurodegenerative diseases without direct evidence may be premature. The manuscript would benefit from emphasizing the need for further research to robustly establish these connections and clarify the nuanced roles of astrocytes in neurodegeneration.

We would like to thank the reviewer for this important correction and comment. We have included the following reference to support the observation of cortical astrocytic domain heterogeneity: Wilhelmsson et al. Redefining the concept of reactive astrocytes as cells that remain within their unique domains upon reaction to injury. PNAS, 2006, 103(46), 17513-17518.

Additionally, we have included a statement in support of further research at the end of the paragraph:

Line 66-67: “Nonetheless, further research is required to clarify if astrocyte region-specific heterogeneity and dysregulation is a driving mechanism in neurodegeneration.”

Lines 115-116:  This statement requires multiple citations, or a single reference to a relevant review paper.

We would like to thank the reviewer for this comment. We have included the following reference to support the statement: Verkhratsky et al. Astrocytes in human central nervous system diseases: a frontier for new therapies. Signal Transduction and Targeted Therapy, 2023, 8(1), 396.

Lines 177-200: While this passage succeeds in providing aggregated evidence of non-cell autonomous astrocytic mechanisms involved in various neurodegenerative diseases, heavy emphasis is placed on AD and FTD relative to the other conditions mentioned (e.g., PD, HD, MS, etc.). To address this, the authors should aim to provide a more balanced overview by dedicating similar lengths of discussion to each disease. This could involve expanding sections on MS, PD, FTD, and ALS to include more detailed descriptions of astrocyte contributions, like the depth provided for AD.

Moreover, the presentation of evidence across different conditions seems fragmented, which might hinder the reader’s ability to follow the complex interplay of astrocyte contributions to these diseases. To improve the organization and readability of the passage, the authors could consider restructuring the information based on thematic or mechanistic categories rather than by disease. For example, sections could be organized around specific astrocyte functions or secretions (e.g., cytokine secretion, chemokine promotion of immune cell infiltration, contributions to protein aggregation, and effects on neuronal and synaptic functions) and then discuss how these mechanisms play out across the different diseases. Alternatively, the authors can simply split the provided evidence into separate paragraphs based on disease, implementing subsection headings if necessary.

We would like to thank the reviewer for this comment. While we appreciate the comment, we fail to see the described imbalance between diseases in this section. The section commences with an outline of various secretome alterations in different diseases, which is subsequently supported by more in-depth examples further down. While the initial few sentences could appear as fragmented information, they are merely an introduction to the widespread secretomic dysregulation observed across many diseases. Further down, we dive into examples of astrocytic molecule secretion consequently affecting other cell types in both MS, AD and FTD. Finally, we devote a large piece of the section to ALS, where substantial information is available.  As we respectfully believe, this reviewer comment is primarily based on a subjective preference and that the proposed alterations will not necessarily enhance the readability nor understanding of the section, we have refrained from applying any alterations.

Lines 251-252: Care should be taken here to avoid implying that neurodegenerative diseases are overwhelmingly or even more commonly characterized by the excess release of the neurotransmitters listed. Both excess release and inhibited release of neurotransmitters have been implicated in the pathology of neurodegenerative diseases, with the specific patterns varying by diseases and within different stages or regions of the brain in the same disease. Please amend the phrasing of this sentence accordingly.

We would like to thank the reviewer for this comment. We have rephrased the sentence as stated below:

Line 252-254: “In pathological conditions, the release of these transmitters from both neurons and astrocytes is dysregulated, and studies show that excess release causes a continuous self-activation leading to cytotoxicity [42,99,112].”

Lines 336-345: While the passage correctly identifies Nrf2 as a critical regulator and its potential therapeutic implications, expanding on how Nrf2 activation in astrocytes specifically influences neuronal survival and disease progression could offer deeper insights – for example, discussing the downstream effects of Nrf2 activation, such as upregulation of antioxidant response element (ARE)-driven genes, would provide a more comprehensive view.

We would like to thank the reviewer for this comment. While we see the point of the reviewer, we believe this addition would be beyond the scope of this paper. The importance of Nrf2 has also recently been reviewed by others (Cuadrado, 2022; Saha et al., 2022; Arslanbaeva and Bisaglia, 2022).

Lines 397-398: This paragraph would benefit from additional detail regarding the mechanism(s) of lysosomal insufficiency. Is the inhibition of lysosomal degradation referenced here related to inadequate lysosomal acidification?

We would like to thank the reviewer for this comment. The referenced studies describe the lysosomal insufficiency as a stress-induced response to the intracellular accumulation of α-synuclein. We have included this information in the paragraph. Additionally, we have extended the paragraph with additional detail on the lysosomal dysfunction observed in PD.

Line 397-403: “This increased inclusion burden aided by overload/stress-induced insufficient lysosomal degradation, further triggers cellular toxicity and reactivity [93,161]. More specifically, α-synuclein is shown to alter the lysosomal morphology, distribution and function by alkalinization and decreased activity of lysosomal proteases in neuronal cells and idiopathic PD brains [196,197], which correlates with similar observations of disrupted lysosomal proteolysis recently observed in astrocytes in an early-onset PD model [198].”

Line 416: Additional information about the function of LRRK2, either here, or in the previous section, would clarify the significance of its mutation and its relation to autophagy dysregulation and impaired astrocytic protein clearance.

We would like to thank the reviewer for this comment. We have included additional information on the significance of LRRK2 in relation to autophagy (please see below).

Line 422-425: “Mutations in LRRK2 are a common cause of genetic PD [205]. LRRK2 contributes among many functions to the phosphorylation of various proteins from the Rab family as well as P62, thereby filling a prominent role in autophagy initiation and vesicle transport [206].”

Figure 2: The authors carefully distinguish between non cell autonomous and cell autonomous mechanisms of astrocytic contributions to various neurodegenerative diseases in the body text – this distinction/categorization should be reflected in this figure.

We would like to thank the reviewer for this comment. Per suggestion of reviewer 2, we have added additional detail to the figure describing the individual cell-autonomous and non-cell autonomous mechanisms. Please see our answer to point 3.

A few grammatical mistakes have been noted throughout the manuscript and the reviewer simply recommends the authors to do a thorough proof read.

The manuscript has been carefully proofread.

Reviewer 2 Report

Comments and Suggestions for Authors

This review offers an interesting approach to the investigation of the roles of astrocytes in neurodegenerative diseases (NDs). By underlying the common mechanisms across various ND groups (such as Parkinson’s disease, Alzheimer’s disease, Huntington’s disease, Amyotrophic Lateral Sclerosis, etc.), the authors provide a practical illustration of the many roles astrocytes can play and offer a clear account of the integrated functioning of the different cell types in the brain in the pathogenesis of very prevalent conditions. Doing so, they shed light on these conditions and, using the most recent findings, they advocate for a nuanced view of the role of activated astrocytes, emphasizing the need to reconsider these activated cells as being parts of a “spectrum” rather than simple categories of “neuroprotective” and “neurotoxic”. Overall, this review constitutes a well-structured and extensive argument in favor of a view of NDs that goes beyond the contribution of neurons - which is too often the focus in the literature – and draws on a large bank of data from various levels of analysis (molecular, histological, clinical).

General concept comments:

The review claims to offer a fresh perspective by providing an overview of the role of astrocytes in neurodegenerative diseases (NDs) based on the underlying mechanisms rather than focusing on the different diseases. However, several previous reviews have already adopted this approach (Ben Haim et al., 2015; Que et al., 2023). The primary limitation of this review is the lack of novelty. While it extensively draws on existing literature, its main results are already summarized in prior reviews, such as Ben Haim et al., (2015).

Improving various aspects of the manuscript would enhance the review:

1.      The “active” properties of astrocytes are not cited in the introduction, contributing to a perception of astrocytes as “passive” cells. For instance, besides neurotransmitter uptake, astrocytes release gliotransmitters through calcium activity. Although this is described later in the article, it is surprising that it is not included when the authors initially outline the primary function of astrocytes.

2.      The conclusion would benefit from more detailed information. It would be particularly valuable to incorporate future perspectives regarding how the highlighted commonalities could be used practically. Otherwise, the statement “what we learn from one disease could potentially be extrapolated to others, ultimately benefitting more patients” might appear unsubstantiated.

The conclusion also seems overly optimistic and does not consider that we are still far from being able to apply these discoveries regarding astroglial correlates of neurodegeneration in curing (or at least alleviating) NDs’ symptoms.

The vicious circle – initiated by neuroinflammation that leads to neuronal death, which in turn creates conditions that further promote inflammation, and so on – is not adequately emphasized. This cycle is an important component of the progression of NDs.

3.      The figures, while useful in some regards, could benefit from more detail. Figure 1 does not seem necessary, as it does not significantly enhance the information provided in the text. Figure 2, intended to detail the astrocytic mechanisms involved in NDs, merely outlines the primary mechanisms by which astrocytes exert their functions without addressing how these functions are impacted in the context of ND, which is the intended purpose of the figure. It would be beneficial to add a figure that discusses the molecular mechanisms involved in the pathogenesis of different NDs, such as recurring signaling pathways, for instance.

4.      Authors could also include a summary table of the main papers involving astrocytes in neurodegeneration.

5.      Among the various roles that astrocytes can play, sleep regulation emerges as a particularly important one. Moreover, in the context of sleep deprivation, astrocyte-mediated inflammation and subsequent neurodegeneration are crucial in the progression of NDs (Que et al., 2023; Li et al., 2023). It would be interesting to add a small paragraph regarding the influence of sleep deprivation – especially considering the prevalence of sleep disorders among NDs patients – on neurodegeneration and the vicious circle of ND progression that it causes: sleep deprivation induces inflammation, which in turn leads to neurodegeneration and disease progression, thereby exacerbating sleep deprivation, and so forth.

Another example not mentioned in the review is the astrocytic implication in AIDS dementia complex, where there is excitotoxicity involving Ca2+-dependent glutamate release from astrocytes (Rossi et al., 2009), which could also be added.

The role of astrocytes in neurodegeneration comprehensively covers the astrocytic mechanisms that cause or result from neurodegeneration. However, it omits the role of astrocytes in the pathogenesis of comorbidities accompanying NDs. For instance, inflammation has been proposed as “a central mechanism underlying the association between AD and most of its comorbidities” (Santiago and Potashkin., 2021). This includes depression, which has been associated with altered astrocytic morphology and function (Nagy et al., 2017; Bellesi et al., 2018; Del Moro et al., 2022).

Specific comments:

1.      “Astrocytes were first described as an individual cell type by Hungarian anatomist and histologist Michael von Lenhossék (Mihàly Lenhossék) in 1895” (lines 27-28)

Even though the name “astrocyte” appeared in 1895, the description of a glial cell type with star-like morphology started, at least, in 1865 with Otto Deiters. Camillo Golgi did another study of this cell type in 1872 and even observed that these cells establish endfeet structures on blood vessels, making them likely intermediate between vessels and neurons (Parpura et al., 2012).

2.      “Astrocytes are crucial for the integrity and function of the neuronal network as they form tripartite synapses with neurons and thereby ensure activity modulation and neurotransmitter regulation through e.g. glutamate uptake.” (lines 41-43)

This sentence seems too weak since astrocytes are required for neuronal survival (Wagner et al., 2006) and since a loss of astrocytic function can drive neurodegeneration (Brenner et al., 2001; Li et al., 2005). It would be easier to make a case for a major role of astrocytes in neurodegeneration if that were established at the onset of the article.

3.      The classification of “Tripartite synapse: Neurotransmitter regulation and synapse function” (line 285) in the “Cell autonomous mechanisms” is misleading, as it involves neurons as well as astrocytes.

4.      No articles are cited to support the claim “Lack of astrocytic neuro-supportive functions likely influences these mechanisms in NDs” (461-462).

5.      “It is fascinating to speculate how many sub-types and sub-states exist, and how astrocytic adaptability might drive neurodegenerative disease progressions” (lines 488-490)

This sentence involves a judgment on the authors’ part and does not add anything scientifically.

Comments on the Quality of English Language

Some errors in grammar and spelling, as well as occasional difficulty in phrasing, can be observed. However, overall, it is readable.

Author Response

We would like to thank the reviewers for their thorough evaluation of our manuscript and for their constructive comments. Their input has without a doubt improved our manuscript. Please find the answers to the individual comments below. In addition to the changes outlined below, we have changed the phrasing “glial scar formation” to glial border formation” to accommodate the updated terminology in the field. All changes have been marked with track-changes in the word-document.

Reviewer 2:

This review offers an interesting approach to the investigation of the roles of astrocytes in neurodegenerative diseases (NDs). By underlying the common mechanisms across various ND groups (such as Parkinson’s disease, Alzheimer’s disease, Huntington’s disease, Amyotrophic Lateral Sclerosis, etc.), the authors provide a practical illustration of the many roles astrocytes can play and offer a clear account of the integrated functioning of the different cell types in the brain in the pathogenesis of very prevalent conditions. Doing so, they shed light on these conditions and, using the most recent findings, they advocate for a nuanced view of the role of activated astrocytes, emphasizing the need to reconsider these activated cells as being parts of a “spectrum” rather than simple categories of “neuroprotective” and “neurotoxic”. Overall, this review constitutes a well-structured and extensive argument in favor of a view of NDs that goes beyond the contribution of neurons - which is too often the focus in the literature – and draws on a large bank of data from various levels of analysis (molecular, histological, clinical).

General concept comments:

The review claims to offer a fresh perspective by providing an overview of the role of astrocytes in neurodegenerative diseases (NDs) based on the underlying mechanisms rather than focusing on the different diseases. However, several previous reviews have already adopted this approach (Ben Haim et al., 2015; Que et al., 2023). The primary limitation of this review is the lack of novelty. While it extensively draws on existing literature, its main results are already summarized in prior reviews, such as Ben Haim et al., (2015).

We would like to thank the reviewer for their overall positive evaluation. We acknowledge that previous authors of reviews have adapted a similar approach, and have therefore changed the wording “fresh perspective” to “different perspective”. While our manuscript shares structural similarities with the reviews mentioned above, our review still provides a different approach compared to contemporary reviews within this field of study.

Improving various aspects of the manuscript would enhance the review:

  1. The “active” properties of astrocytes are not cited in the introduction, contributing to a perception of astrocytes as “passive” cells. For instance, besides neurotransmitter uptake, astrocytes release gliotransmitters through calcium activity. Although this is described later in the article, it is surprising that it is not included when the authors initially outline the primary function of astrocytes.

 We would like to thank the reviewer for this comment. As pointed out by the reviewer, this important astrocyte function is described in more detail in section 4.1 Communication. To avoid too much repetition, we have refrained from describing every astrocytic function in the introduction, but focused on introducing these in their individual sections.

  1. The conclusion would benefit from more detailed information. It would be particularly valuable to incorporate future perspectives regarding how the highlighted commonalities could be used practically. Otherwise, the statement “what we learn from one disease could potentially be extrapolated to others, ultimately benefitting more patients” might appear unsubstantiated.

The conclusion also seems overly optimistic and does not consider that we are still far from being able to apply these discoveries regarding astroglial correlates of neurodegeneration in curing (or at least alleviating) NDs’ symptoms.

The vicious circle – initiated by neuroinflammation that leads to neuronal death, which in turn creates conditions that further promote inflammation, and so on – is not adequately emphasized. This cycle is an important component of the progression of NDs.

We would like to thank the reviewer for these comments. Verkhratsky and colleagues give a very nice overview of future perspectives regarding astrocyte targeted therapy (Verkratsky et al., 2023), which is the reason, we have refrained from including this information in our review conclusion. Nonetheless, we agree with the reviewer that our conclusion is very optimistic, so we have included a larger emphasis on the need for further studies to elucidate the mechanistic overlap across diseases.

Line 498-512: “Astrocyte physiology and pathology is complex. It is fascinating to speculate how many sub-types and sub-states exist, and how astrocytic adaptability might drive neurodegenerative disease progressions. Despite their profound heterogeneity, astrocytes as a cell group share common pathological characteristics across a broad spectrum of neurodegeneration. Here, we have highlighted the disease commonalities in some of the most common mechanisms, but many more are likely present, and further research is undoubtedly warranted. What we learn from one disease could potentially be extrapolated to others, ultimately benefitting more patients. Current treatment regimens largely follow the dominant field of neurocentric studies, but this neuronal favoring has unfortunately mounted in many unsuccessful clinical trials. Given the paramount role of astrocytes in ensuring optimal neuronal function and survival, these glial cells constitute important therapeutic targets in drug development. With this review, we encourage future studies in astrocyte mechanisms with a cross-disease and cross-model investigative approach in an effort to lower the variability and potentially disclose new shared therapeutic targets for multiple neurodegenerative diseases.”

  1. The figures, while useful in some regards, could benefit from more detail. Figure 1 does not seem necessary, as it does not significantly enhance the information provided in the text. Figure 2, intended to detail the astrocytic mechanisms involved in NDs, merely outlines the primary mechanisms by which astrocytes exert their functions without addressing how these functions are impacted in the context of ND, which is the intended purpose of the figure. It would be beneficial to add a figure that discusses the molecular mechanisms involved in the pathogenesis of different NDs, such as recurring signaling pathways, for instance.

 We would like to thank the reviewer for these comments and suggestions. We have modified figure 2, which now also provides a bullet point list of altered pathological mechanisms described in this review (please see below). Additionally while this review was under referee assessment, Liddelow and colleagues published a review with an interesting overview of intrinsic signalling pathways and transcriptional regulators of astrocyte reactivity (Liddelow et al., 2024), which nicely accommodates our manuscript.

Updated  Figure 2 can be found in the attached file

Figure 2. Common mechanisms involved in astrocytic pathology in neurodegenerative diseases. Figure is created with biorender.com

  1. Authors could also include a summary table of the main papers involving astrocytes in neurodegeneration.

 We would like to thank the reviewer for this suggestion. We initially attempted to include a table with an overview of the various model systems used in assessing the different disease mechanisms. Unfortunately, due to the vast amount of included references such a table became unmanageable. In our revised figure 2, we now give a more detailed summary of the currently known mechanisms, which we believe renders a table redundant.

  1. Among the various roles that astrocytes can play, sleep regulation emerges as a particularly important one. Moreover, in the context of sleep deprivation, astrocyte-mediated inflammation and subsequent neurodegeneration are crucial in the progression of NDs (Que et al., 2023; Li et al., 2023). It would be interesting to add a small paragraph regarding the influence of sleep deprivation – especially considering the prevalence of sleep disorders among NDs patients – on neurodegeneration and the vicious circle of ND progression that it causes: sleep deprivation induces inflammation, which in turn leads to neurodegeneration and disease progression, thereby exacerbating sleep deprivation, and so forth.

Another example not mentioned in the review is the astrocytic implication in AIDS dementia complex, where there is excitotoxicity involving Ca2+-dependent glutamate release from astrocytes (Rossi et al., 2009), which could also be added.

The role of astrocytes in neurodegeneration comprehensively covers the astrocytic mechanisms that cause or result from neurodegeneration. However, it omits the role of astrocytes in the pathogenesis of comorbidities accompanying NDs. For instance, inflammation has been proposed as “a central mechanism underlying the association between AD and most of its comorbidities” (Santiago and Potashkin., 2021). This includes depression, which has been associated with altered astrocytic morphology and function (Nagy et al., 2017; Bellesi et al., 2018; Del Moro et al., 2022).

We would like to thank the reviewer for this comment. While we see the point of the reviewer, we believe these additions would be beyond the scope of this paper. The importance of astrocytes in sleep deprivation, AIDS dementia and comorbidities are also described by others in recent reviews mentioned above (Que et al. 2023, Li et al. 2023 ), as well as in Tice et al. 2020, Wahl and Al-Harthi 2023, Gonzalez 2021, and Verkhratsky et al, 2023.

Specific comments:

  1. “Astrocytes were first described as an individual cell type by Hungarian anatomist and histologist Michael von Lenhossék (Mihàly Lenhossék) in 1895” (lines 27-28)

Even though the name “astrocyte” appeared in 1895, the description of a glial cell type with star-like morphology started, at least, in 1865 with Otto Deiters. Camillo Golgi did another study of this cell type in 1872 and even observed that these cells establish endfeet structures on blood vessels, making them likely intermediate between vessels and neurons (Parpura et al., 2012).

We would like to thank the reviewer for this comment. We have changed the wording of the sentence to better convey our message.

Line 27-29: “Astrocytes received their name from Hungarian anatomist and histologist Michael von Lenhossék (Mihàly Lenhossék) in 1895 due to their mesmerizing star-like morphology [1].”

  1. “Astrocytes are crucial for the integrity and function of the neuronal network as they form tripartite synapses with neurons and thereby ensure activity modulation and neurotransmitter regulation through e.g. glutamate uptake.” (lines 41-43)

This sentence seems too weak since astrocytes are required for neuronal survival (Wagner et al., 2006) and since a loss of astrocytic function can drive neurodegeneration (Brenner et al., 2001; Li et al., 2005). It would be easier to make a case for a major role of astrocytes in neurodegeneration if that were established at the onset of the article.

We would like to thank the reviewer for this comment. We agree with the reviewer that astrocytes are important for neuronal survival, however the section is merely intended to give a small introduction to various astrocytic functions in the CNS under physiological conditions.

  1. The classification of “Tripartite synapse: Neurotransmitter regulation and synapse function” (line 285) in the “Cell autonomous mechanisms” is misleading, as it involves neurons as well as astrocytes.

We would like to thank the reviewer for this comment. We agree that neurons are involved in the mechanisms, however we believe their mention doesn’t dispute the cell autonomous involvement of astrocytes.

  1. No articles are cited to support the claim “Lack of astrocytic neuro-supportive functions likely influences these mechanisms in NDs” (461-462).

We would like to thank the reviewer for this comment. We have included the following reference to support the statement: Verkhratsky et al. Astrocytes in human central nervous system diseases: a frontier for new therapies. Signal Transduction and Targeted Therapy, 2023, 8(1), 396.

  1. “It is fascinating to speculate how many sub-types and sub-states exist, and how astrocytic adaptability might drive neurodegenerative disease progressions” (lines 488-490)

This sentence involves a judgment on the authors’ part and does not add anything scientifically.

We would like to thank the reviewer for this comment. As described above, our conclusion highlights the overall aim of this review, which is to encourage further cross-disease studies with an emphasis on disease mechanisms in astrocytes. This encouragement is very much driven by the enthusiasm that we and others share for the complexity of astrocytes, which is represented in this sentence. 

Comments on the Quality of English Language

Some errors in grammar and spelling, as well as occasional difficulty in phrasing, can be observed. However, overall, it is readable.

The manuscript has been carefully proofread.
